# The effects of plyometric training with speed and weight overloads on volleyball players' strength, power, and jumping performance

**Ahmad Reza Iranpour[1], Mohammad Hemmatinafar**◉[1]*, **Javad Nemati[1], Mohsen Salesi[1], Hamed Esmaeili[2], Babak Imanian[1]**

**1** Department of Sport Sciences, Faculty of Education and Psychology, Shiraz University, Shiraz, Iran,
**2** Department of Sport Injuries and Corrective Exercise, Faculty of Sport Sciences, University of Isfahan, Isfahan, Iran

* m.hemmatinafar@shirazu.ac.ir

## Abstract

### Background

The principle of gradual overload is crucial in improving sports performance, yet the effects of combining speed and weight overloads in plyometric training remain understudied.

### Objective

This study investigated the effects of plyometric training with speed and weight overloads on isokinetic strength, explosive power, and agility in volleyball players.

### Method

Forty male volleyball players were randomly assigned to four groups: plyometric training (PT), plyometric training with speed overload (PTS), plyometric training with weight overload (PTW), and plyometric training with both speed and weight overload (PTSW). Each group completed a four-week plyometric training regimen. Performance metrics, including Sarjent's jump height (SJH), Spike jump height (SPJH), Sheppard test (ShT), muscle voluntary isometric contraction (MVIC), rate of force development (RFD), absolute peak torque (PTQ), relative peak torque (RPT), and average power (AP) of knee extensors and flexors were measured before and after the intervention.

### Results

SJH improved significantly in PTS (P = 0.012), PTW (P = 0.041), and PTSW (P = 0.001) compared to PT. SPJH showed substantial gains in PTS (P = 0.002), PTW (P = 0.001), and PTSW (P = 0.001) compared to PT. Average jump height and highest jump in ShT were also significantly higher in PTS, PTW, and PTSW (P < 0.05). Additionally, RFDext240˚/s was enhanced considerably in PTS (P = 0.001) and PTSW (P = 0.001).

**Data Availability Statement:** All relevant data are within the manuscript and its Supporting Information files.

**Funding:** The author(s) received no specific funding for this work.

**Competing interests:** The authors have declared that no competing interests exist.

## Conclusion

Based on the results, plyometric training with speed and weight overloads (PTSW) demonstrated superior enhancements in isokinetic strength, explosive power, and jumping performance. This combined approach is highly effective and significantly benefits male volleyball players, aiming to enhance their physical abilities.

## 1. Introduction

Volleyball is characterized by dynamic, explosive, interval-based movements that require technical skill and tactical knowledge, and better performers play with a "higher attitude" from both a talent and vertical jump performance perspective [1]. In volleyball, the level of physical fitness of the players is primarily related to the results of the teams in the competitions [2]. The unique characteristics of volleyball include speed, maximum vertical jump, frequent changes of direction, and high and overhead movements [1]. Volleyball players make about 30,000 to 40,000 jumps per year [3]. In addition, these athletes perform more than 100 high-intensity jumps in each competition [3]. Regarding biomechanics, spike height, speed, and ball path are the main variables that determine the success rate of spike in volleyball [4]. Jumping ability covers almost half of the entire game, which shows the need for planning to develop lower body strength and power [1].

Power, agility, and speed are essential physical fitness factors in volleyball players' performance [5]. Considering that power is influenced by two factors, strength and speed, and speed is also an essential and influential factor in agility, improving and investigating and using speed as a variable in the training of volleyball athletes can probably affect power, agility, and it will affect the performance of volleyball athletes [5]. Speed is a person's ability to move all or parts of the body in space in the shortest possible time without determining the direction [5]. Considering that during jumping, the driving force for jumping is limited (the anatomy of the length of the leg and its joints), more work (force production) is done on the center of gravity by shortening the time of the excitation phase [6]. This means that the average execution speed will be higher because the speed is equal to displacement divided by time, to increase the speed, the desired removal must be done in a shorter time [6]. Additionally, we can apply more force at the moment by shortening the excitation time and increasing the speed, according to the formula: $F \times s = \frac{1}{2} \times m \times v^2$ [6]. As a result, the height of the jump will likely increase [6]. In this regard, Sheppard et al. (2011) used an elastic band hanging from the ceiling to investigate the effect of over-speeding (assisted jump) on elite volleyball players for five weeks [7]. Compared to traditional jump training, the results showed that assisted jump training has increased the shortening speed of extensor muscles and ultimately improved vertical jump [7]. So far, speed has been used in various ways as a variable in training design. These methods have aligned with the development of speed, power, and different functional factors. Among the researchers that have used speed as a variable in training design, there is velocity-based training (VBT), which is a type of training that uses speed as an intensity factor and with the approach of developing strength, power development, functional training and injury prevention is provided [8], Miller et al. (2020) have used speed as a variable in training by changing the weight of the baseball bat (decreasing and increase the weight compared to the usual weight) [9]. One of the other methods of using the speed variable is overspeed training (OST). This method has been used for years as a particular exercise for athletics, which is given to the athlete at a speed higher than the maximum speed by using tilt, drag, and towing in the

direction of movement. The resulting adaptations have led to the improvement of the frequency, stride length, and, finally, the improvement of the athlete's maximum speed [10]; this method is also called assisted training. This method has also been used in designing strength exercises and improving jump height, called assisted jumping [7, 11]. Assisted jumping often increases athletes' power output and vertical jump performance. Overspeed training increases the power of the lower limbs [12]. This type improves jumping ability by reducing the effective mass of a jumper and increasing the peak acceleration during the jump due to reduced load conditions [13].

Another undeniable principle in developing sports performance and adaptation is the principle of gradual overload [14]. The overload principle states that a disturbance in the body's homeostasis, including cells, tissues, and organs, is necessary for effective exercise adaptation [14]. Applying overload in training is based on the goals and unique needs of movement and is done according to other training principles, environment, timing, and facilities [15]. There are many strategies to improve power, speed, agility, and strength performance. Regardless of the type of exercise that enhances these factors, paying attention to exercise principles will lead to better progress and reduce injuries [15]. The principle of overload (exposure of tissues to more stress than usual) is one of the principles of exercise science [15]. Three essential things to consider when maximizing output power [16]. First, trainers should consider the overall increase in muscle strength due to its direct relationship with the level of strength development and output power [16]. The second case is the importance of developing the ability to produce a lot of force in short periods, expressed by the rate of force development (RFD) [16]. Also, the evaluation of isokinetic strength in volleyball players is used to evaluate performance and predict injury [17, 18]. Additionally, there is strong evidence that eccentric strength of the knee/hip flexors and extensors is essential for optimal deceleration during change-of-direction (COD) maneuvers [19–21]. However, new evidence suggests that in addition to eccentric strength, concentric forces of knee extensors and flexors also play an essential role in deceleration and COD [22]. For example, in elite youth soccer players, Kadlubowski et al. (2021) have shown that COD is influenced by concentric power and maximal strength [22]. Another study has also confirmed that the development of concentric strength of the knee extensors and flexors is necessary to improve performance in the COD and deceleration in youth soccer players [23]. Finally, the third thing is the ability to produce a lot of force with an increase in shortening speed, which leads to an improvement in power production and an increase in vertical jump height. The intense interaction between these three items is essential in producing high power. Support for the mutual relationship between maximum strength, speed of force development, and maximum power output has been seen in scientific literature, and a high correlation has been found between these variables [24].

Considering the above, plyometric training is integral to volleyball players' training. However, the effect of plyometric training with velocity and weight overload on volleyball athletes has not been entirely determined, and contradictory results are available. Therefore, this research aimed to investigate the application of velocity and weight overload on volleyball athletes' isokinetic strength, power, and agility.

## 2. Materials and methods

### 2.1 Participants

In this study, forty male volleyball players with approximately three years of experience in the Iran Volleyball League participated. The demographic information of the participants is listed in Table 1. Participants had no known diseases or medical issues and did not consume supplements or medications. Furthermore, the participants did not smoke or drink alcohol or

**Table 1. The anthropometric data of participants.**

| Characteristic | Mean ± SD (n = 40) |
|---|---|
| Age (years) | 23.65 ± 2.76 |
| Height (cm) | 188.1 ± 6.75 |
| Weight (kg) | 78.26 ± 8.44 |
| BMI (kg/m$^2$) | 22.1 ± 2.09 |

caffeinated beverages during data collection [25]. The current study was reviewed and approved by The Human Research Ethics Committee of Shiraz University (ethics approval code: IR.US.PSYEDU.REC.1402.101. 2023) and carried out in accordance with the Declaration of Helsinki. According to the above conditions, all forty participants were selected from the volleyball players who volunteered to participate in this study. It should be noted that there was no withdrawal, and all the subjects participated in the pre-test and post-test. Additionally, all participants were members of the same training camp, and their training regime was the same under the supervision of trainers.

## 2.2 Sample size calculation

The sample size was determined based on estimating the differences in effect sizes expected from the training protocols. Previous research on the effects of various speed training protocols on sprint acceleration kinematics and muscle strength and power [26], indicated a large effect size (0.8) for this study. Utilizing G*Power 3.1.9.2, with a confidence interval of 95% and an analysis power of 0.80, and assuming an effect size of 0.8 at an alpha level of 0.05, it was determined that a minimum of 24 participants (6 participants per group) are required for this study. Therefore, 40 participants were selected for the current study (10 participants for each group) [27].

## 2.3 Study design

This study was designed as a randomized, double-blinded, pre-and post-test experiment (Fig 1). The participants' general health was assessed by a general practitioner through a medical examination, alongside completing a physical activity readiness questionnaire. Written informed consent was obtained from all participants, including detailed explanations of the study protocol, benefits, potential risks, and possible complications. Subsequently, the participants were randomly assigned to one of four groups, each consisting of 10 individuals: plyometric training (PT), plyometric training with speed overload (PTS), plyometric training with weight overload (PTW), and plyometric training with both speed and weight overload (PTSW) (Fig 1). Over four consecutive days, the participants attended the laboratory at Esfahan University. After completing a standardized warm-up, they underwent isokinetic and isometric strength testing using the Biodex Isokinetic Dynamometer (Biodex System 4 Pro, USA). The isokinetic tests measured maximum torque at angular velocities of 120˚/s and 240˚/s with concentric-concentric contraction, while the isometric tests were conducted at a fixed knee angle of 45˚. On a separate day, the participants performed additional performance tests, including the Sargent's jump test to measure maximum vertical jump height, the spike jump test to assess volleyball-specific jump height, the agility T-test, and the volleyball functional Sheppard test, also referred to as the repeated effort test. Each group followed a specific plyometric training regimen over four weeks, as detailed in Table 2, with variations based on the inclusion of speed and/or weight overloads. Forty-eight hours after the final training session, the participants underwent post-tests, replicating the same procedures used during the pre-

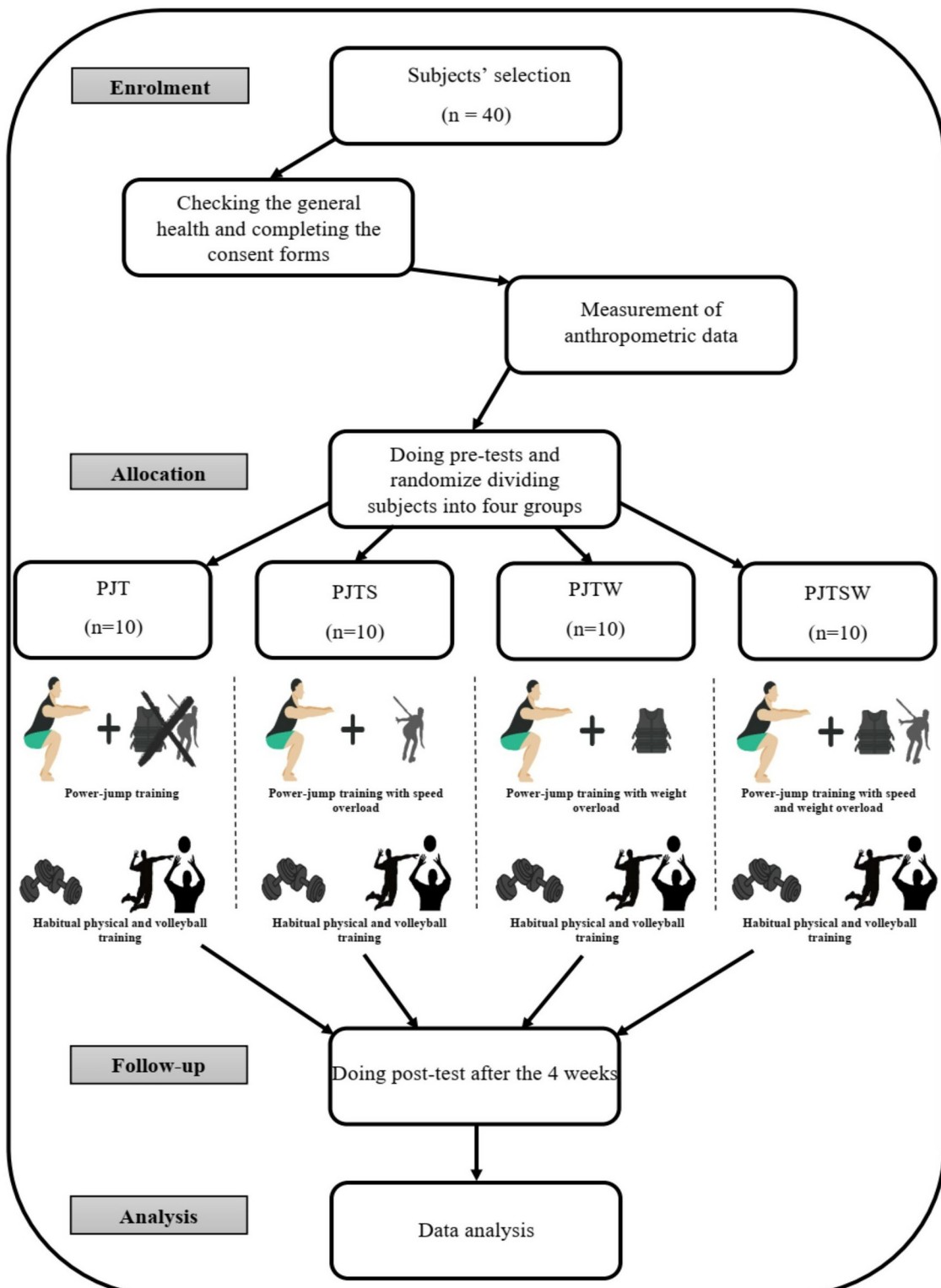

**Fig 1. Flowchart illustrating the different phases of the research and study selection.** PT: plyometric training, PTS: plyometric training with speed overload, PTW: plyometric training with weight overload, PTSW: plyometric training with speed and weight overload.

**Table 2. Plyometric training (PT) protocol for the four weeks in the four groups of the study.**

|  | First week (2 sessions) (set × rep) | Second week (3 sessions) (set × rep) | Third week (2 sessions) (set × rep) | Fourth week (3 sessions) (set × rep) |
|---|---|---|---|---|
| Squat jump | 4×5 | 4×4 | 4×6 | 4×5 |
| Lateral jump | 4×5 | 4×4 | 4×6 | 4×5 |
| Lunge jump | 4×5 | 4×4 | 4×6 | 4×5 |
| Depth jump | 4×5 | 4×4 | 4×6 | 4×5 |
| Pogo jump | 4×6 | 4×6 | 4×8 | 4×6 |
| Spike volleyball jump | 4×5 | 4×4 | 4×6 | 4×5 |
| Number of Jump per session | 124 | 104 | 152 | 124 |
| Number of Jump per week | 248 | 312 | 304 | 372 |

set: Sets, rep: Repetition

test phase. These post-tests were conducted to evaluate the outcomes of the intervention. To ensure consistency, all participants were given a standardized breakfast containing 250 kcal (45 g carbohydrates, 9 g protein, and 5 g fat) one hour and thirty minutes before each testing session [25]. Both pre-test and post-test sessions were conducted simultaneously, between 8:30 AM and 1:00 PM, to minimize any potential effects of diurnal variations. During the trials, participants were allowed to drink water freely as needed.

## 2.4 Training protocols

The power jump training protocol started after grouping the athletes and performing the pre-tests. The participants underwent ten training sessions over four weeks. All study groups performed the power jump training protocol throughout the training program, with or without weight or velocity overload. The training schedule included two sessions in the first and third weeks and three in the second and fourth weeks. According to Table 2, the PT group performed the protocol without any weight or velocity overload, and the PTS group performed the same protocol with velocity overload. The PTW group performed the same protocol with weight overload, and the PTSW group did the protocol with velocity and weight overload similarly. At least 48 hours of recovery were considered for each training session. The number of jumps per session and week was the same for all groups during four weeks. Moreover, in all training groups, athletes were instructed to perform the exercise with maximum power and explosiveness, according to RPE≥17 [28].

## 2.5 Plyometric training group (PT)

The group known as PT underwent a power jump training protocol without weight or velocity overload. The training schedule consisted of two sessions in the first and third weeks and three in the second and fourth weeks. The details of the protocol are in Table 2.

## 2.6 Plyometric training with speed overload group (PTS)

In the PTS group, An adjustable non-elastic band was suspended from the high metal structures of the ceiling. Then, a 32-millimeter single-layer power loop resistance band was attached to the hanging scale and non-elastic band (Fig 2). The height of the non-elastic band was adjusted according to the athlete's height so that it reached a certain weight at the knee jump angle of 120 degrees. Then, the hanging scale was removed, and the training began. This load

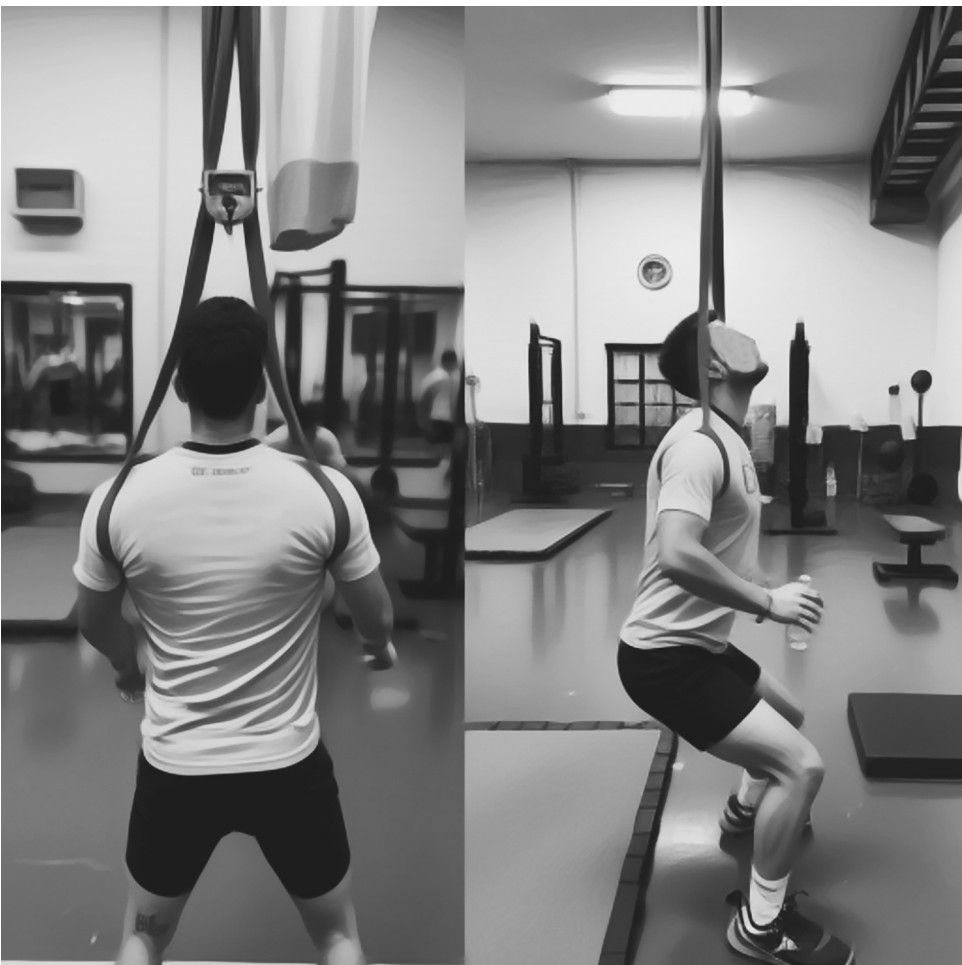

**Fig 2. The method of applying speed overload.**

reduction was 20% of body weight in the first week, 15% in the second week, and 10% of body weight in the third and fourth weeks [12, 29, 30]. The protocol was performed with the following Table 2.

## 2.7 Plyometric training with weight overload group (PTW)

Barbells, weighted vests, plates, and elastic bands were used in the plyometric training with the weight overload group. The basis of applying overload was the one repetition maximum (1RM) squat calculated from the Brzycki formula [weight lifted / [1.0278 −(0.0278 × repetitions performed)] [31, 32]. In the first week, 20% of 1RM, 15% of 1RM in the second week, and 10% of 1RM in the third and fourth weeks were overloaded. The exercises were performed under Table 2.

## 2.8 Plyometric training with speed and weight overload group (PTSW)

In this group, both speed and weight overload protocols were applied. The first two sets were performed with weight overload (precisely according to the weight overload method), and the following two sets were performed with speed overload (precisely according to the speed overload method). The exercises were performed under Table 2.

**2.8.1 Sarjent's jump test.** The Sargent's jump test was used to evaluate the Sargent jump Height (SJH). The participants chalk the end of their fingertips. Then, stand beside the wall, keeping both feet on the ground. From a static position, the participants jump as high as possible and mark the wall with the chalk on their fingers. All evaluations were performed three times (with a one-minute passive rest between them), and the best score was considered the final result [33].

**2.8.2 Spike jump test.** It is a jump that volleyball players perform by performing three volleyball steps, swinging their arms, and aiming to hit the spike at the highest possible height. It is known as a performance test in volleyball players by measuring the height of the jump in evaluations (SPJH) [34]. All evaluations were performed three times, separated by one minute of rest, and the best score was used as the final result.

**2.8.3 Agility test.** T-test is used in agility evaluation. Subjects were asked to sprint forwards 9.14 m from the start line to the first cone and touch the tip with their right hand, shuffle 4.57 m left to the second cone and touch with their left hand, then shuffle 9.14 m to the right to the third cone and touch with their right, shuffle 4.57 m back left to the middle cone and touch with their left hand before finally backpedaling to the start line. Time begins upon subjects passing through the timing gates and stops upon them passing through on return. The test will not be counted if the subject crosses one foot in front of the other while shuffling, fails to touch the base of the cones, or fails to face forward throughout the test. The best time of three successful trials was recorded [35].

**2.8.4 Isokinetic strength parameters.** During the isokinetic assessments, participants were seated on the Biodex Isokinetic Dynamometer (Biodex System 4 Pro, USA) with their hips and knees positioned at a 90-degree angle. The dynamometer chair and lever arm were adjusted to align the knee joint axis with the dynamometer's rotational axis. Participants were secured using adjustable straps across the torso, hips, and thigh to minimize extraneous movements. The ankle strap was positioned just above the medial malleolus for consistent force application [36, 37]. The concentric isokinetic strength of knee extensors (ext) and flexors (flx) (dominant leg) was evaluated at two different speeds: 120˚/s and 240˚/s. Absolute peak torque (PTQ), relative peak torque (peak torque per body weight) (RPT), and Average power (AP) were measured during five repetitions (PTQflx120˚/s, PTQext120˚/s, PTQflx240˚/s, PTQext240˚/s, RPTflx120˚/s, RPText120˚/s, RPTflx240˚/s, RPText240˚/s, APflx120˚/s, APext120˚/s, APflx240˚/s, APext240˚/s). Also, the average rate of force development (RFD) was calculated using the absolute peak torque/time to peak torque equation (RFDflx120˚/s, RFDext120˚/s, RFDflx240˚/s, RFDext240˚/s) [37, 38].

**2.8.5 Isometric strength parameter.** Participants were seated in the same position for the isometric tests as for the isokinetic assessments. The knee joint was fixed at a 45-degree angle using the locking mechanism of the Biodex System 4 Pro Dynamometer (Biodex Medical Systems, USA). Straps were applied across the upper body, hips, and tested limb to ensure stability and prevent compensatory movements. Participants were instructed to maintain a neutral spine posture throughout the test and were verbally encouraged to exert maximal effort during all trials [38]. The assessment of Muscle Voluntary Isometric Contraction (MVIC) strength focused on the dominant leg's knee flexor (flx) and extensor (ext) muscles. Each participant performed five maximal voluntary contractions, each lasting 5 seconds, with the knee joint held at a 45-degree angle. The MVIC values for both flexor (MVICflx45˚) and extensor (MVICext45˚) muscles were recorded for analysis [36].

To ensure consistency, each subject's positioning was double-checked before each test session. The equipment was calibrated according to the manufacturer's guidelines before the commencement of testing (Biodex User Manual, 2021).

**2.8.6 Volleyball functional test (Sheppard).** The Sheppard repeat-effort test evaluated volleyball performance (Right-Hand Version) (Fig 3). A vertical jump and reach device are set up 4.9 feet back (1.5 m) from the net and 4.9 feet from the left sideline. Two-timing light setups are used, with one being placed 4.9 feet (1.5 m) from the sideline, just inside of the vertical jump (VJ) and reach device; the other is set up perpendicular to the first and is placed on the 10-feet line (3 m). The adjustable blocking apparatus is set up on the opposite side of the net in the middle of the court, with the center of the apparatus at 15 feet (4.5 m). The test begins at the start line, where the athlete performs their first spike jump (SPJ) (action 1), which is to be registered on the VJ and reach the device (recorded, and the vanes must be reset quickly). The athlete lands near the net and initiates movement through the first set of timing lights (action 2) toward the block jump (BJ) task. The athlete then performs a BJ at each mounted volleyball (actions 3 and 4), and an error is recorded against the movement time if both hands do not contact the ball. On completing the blocking task, the athlete moves laterally toward the far sideline (action 6) until their right foot either touches or goes beyond the line setup 3.3 feet (1 m) from the sideline. The athlete then reverses the process (actions 7–10) and, on completing the last blocking task, uses transition footwork and an open step consistent with volleyball movement to move backward and diagonally through the second set of timing lights, rounding a 6-feet 6-inch (2 m) high marker to complete a final SPJ (action 11). This final SPJ is the final action of the first repetition, and four repetitions are performed to complete the test [39] (Fig 3). It should be noted that if the participant deviated from the protocol, the test was repeated with an interval of 15 minutes (for recovery). The average time of doing the Sheppard test (average time of ShT), the shortest time of doing the Sheppard test (shortest time of ShT), the average jump height of the Sheppard test (average jump height of ShT), and the maximum

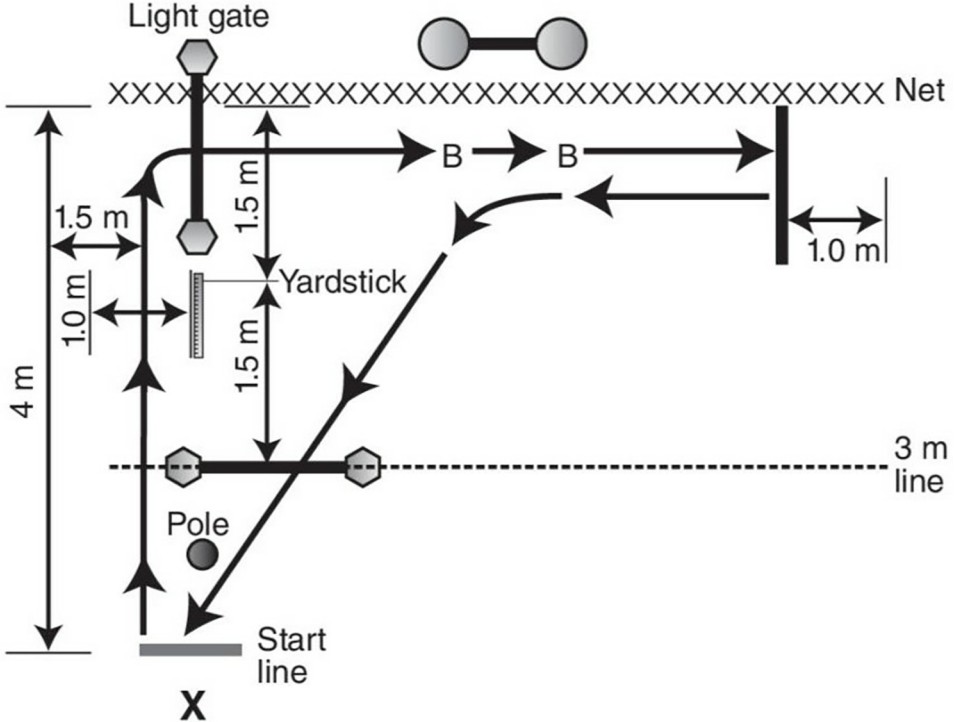

**Fig 3. Schematic design of Sheppard repeat-effort test [83].**

height of the jump during the Sheppard test (highest jump of ShT) were recorded as the participants' scores.

## 2.9 Statistical analyses

All data were analyzed using both descriptive and inferential statistical methods. The normality of data distribution was assessed using the Shapiro-Wilk test. Repeated measures analysis of variance (ANOVA) was employed to evaluate the main effects of the pre-test and post-test on the measured indicators, with pairwise differences determined through the Bonferroni post-hoc test, incorporating Syntax codes. Statistical analyses were performed using SPSS software (version 28, IBM-SPSS Inc., Chicago, IL, USA). Partial eta squared (pEta$^2$) was calculated to measure effect size for interaction and main effects. Based on Cohen's guidelines, pEta$^2$ values of $\geq$0.01 indicate small effects, $\geq$0.059 indicate medium effects, and $\geq$0.138 indicate large effects [40]. Statistical significance was set at P $\leq$ 0.05, and results are presented as mean ± standard deviation (SD). Figures were generated using GraphPad Prism software (version 8.4.3, GraphPad Software, San Diego, CA, USA).

## 3. Results

Table 1 presents descriptive statistics, including the mean and standard deviation for the measured variables. Additionally, Table 3 outlines these variables' mean and standard deviation for both the pre-test and post-test assessments. The Shapiro-Wilk test results indicate that the data exhibit a normal distribution (P > 0.05).

### 3.1 Functional tests results

Statistical analysis revealed a significant main effect on agility ($F_{1.00}$ = 10.67, P = 0.002, pEta$^2$ = 0.229) (Table 4). The Bonferroni posthoc test showed significant improvements in agility for the PTS (P = 0.007) and PTSW (P = 0.047) groups in the post-test compared to the pre-test. In contrast, no significant differences were observed for the PT (P = 0.908) and PTW (P = 0.143) groups. Additionally, in the post-test, agility improved significantly in PTSW compared to PT (P = 0.047), while other group comparisons were not significant (P > 0.05). There were no significant differences between groups in the pre-test (P > 0.05) (Fig 4).

The main effect on Sarjent's Jump Height (SJH) was also significant ($F_{1.00}$ = 24.02, P = 0.001, pEta$^2$ = 0.400) (Table 4). Post-hoc analysis showed substantial increases in SJH for PTS (P = 0.012), PTW (P = 0.041), and PTSW (P = 0.001) in the post-test compared to the pre-test, while no significant difference was observed for PT (P = 0.877). Furthermore, SJH improved significantly in PTSW compared to PT in the post-test (P = 0.036), with no significant differences among other group comparisons. No significant differences were found between groups in the pre-test (P > 0.05) (Fig 4).

For Spike Jump Height (SPJH), the repeated measures analysis indicated a significant main effect ($F_{1.00}$ = 40.73, P = 0.001, pEta$^2$ = 0.531) (Table 4). SPJH increased significantly in PTS (P = 0.002), PTW (P = 0.001), and PTSW (P = 0.001) in the post-test compared to the pre-test, with no significant difference for PT (P = 0.732). In the post-test, PTSW significantly outperformed PT (P = 0.022), while no other between-group differences were observed. Pre-test results showed no significant differences among the groups (P > 0.05) (Fig 4).

The analysis of the average jump height of the Sheppard Test (ShT) revealed a significant main effect ($F_{1.00}$ = 39.60, P = 0.001, pEta$^2$ = 0.524) (Table 4). Improvements were substantial for PTS (P = 0.001), PTW (P = 0.003), and PTSW (P = 0.001) in the post-test compared to the pre-test, with no significant difference for PT (P = 0.938). No significant differences were observed between groups in the pre-test or post-test (P > 0.05) (Fig 4).

Table 3. Means and standard deviations (Mean ± SD) of measured variables.

| | Pre-test | | | | Post-test | | | |
|---|---|---|---|---|---|---|---|---|
| | PT (n = 10) | PTS (n = 10) | PTW (n = 10) | PTSW (n = 10) | PT (n = 10) | PTS (n = 10) | PTW (n = 10) | PTSW (n = 10) |
| Agility (s) | 10.33 ± 0.46 | 9.99 ± 0.47 | 10.37 ± 0.63 | 9.87 ± 0.70 | 10.32 ± 0.43 | 9.72 ± 0.50 | 10.23 ± 0.56 | 9.68 ± 0.58 |
| SJH (cm) | 53.70 ± 4.78 | 56.30 ± 3.65 | 55.80 ± 6.56 | 57.40 ± 7.16 | 53.80 ± 4.58 | 58.00 ± 4.18 | 57.10 ± 5.52 | 60.60 ± 6.27 |
| SPJH (cm) | 59.90 ± 5.30 | 62.20 ± 4.58 | 62.30 ± 7.27 | 64.30 ± 7.22 | 59.70 ± 5.05 | 64.10 ± 4.43 | 64.60 ± 5.96 | 67.70 ± 7.21 |
| Average time of ShT (s) | 8.03 ± 0.71 | 7.87 ± 0.50 | 8.06 ± 0.69 | 8.08 ± 0.70 | 8.02 ± 0.70 | 7.87 ± 0.51 | 7.97 ± 0.67 | 8.13 ± 0.58 |
| Shortest time of ShT (s) | 7.78 ± 0.70 | 7.65 ± 0.48 | 7.78 ± 0.72 | 7.75 ± 0.69 | 7.71 ± 0.68 | 7.66 ± 0.51 | 7.69 ± 0.67 | 7.83 ± 0.59 |
| Average jump height of ShT (cm) | 298.90±8.96 | 305.20±7.34 | 300.80±9.99 | 307.90±13.38 | 298.80±9.05 | 307.30±6.96 | 302.60±10.18 | 311.30±13.83 |
| Highest jump of ShT (cm) | 300.40±9.24 | 307.50±7.59 | 302.50±10.18 | 310.60±13.96 | 300.40±9.05 | 309.40±7.57 | 304.70±9.90 | 314.00±13.85 |
| MVICext45˚ (Nm) | 267.60±48.76 | 301.48±42.76 | 262.30±50.17 | 275.95±48.77 | 267.00±41.35 | 307.80±53.02 | 272.40±43.84 | 297.90±42.46 |
| MVICflx45˚ (Nm) | 132.70±16.34 | 147.60±12.42 | 128.80±30.56 | 146.15±17.93 | 131.20±9.69 | 150.80±17.47 | 139.60±23.44 | 154.20±21.93 |
| RFDext120˚/s (Nm/mS) | 836.40±117.67 | 925.30±109.63 | 804.70±148.08 | 947.30±187.78 | 835.40 ±123.83 | 1055.30 ±213.12 | 914.90±202.38 | 1072.90 ±165.86 |
| RFDflx120˚/s (Nm/mS) | 443.30±71.79 | 410.50±141.11 | 401.90±123.02 | 486.30±85.06 | 462.40 ±104.80 | 437.20±125.75 | 491.80±116.14 | 584.50±175.39 |
| RFDext240˚/s (Nm/mS) | 1005.10 ±199.89 | 1154.40 ±130.92 | 1054.90 ±243.02 | 1092.30 ±173.84 | 972.90 ±167.44 | 1296.40 ±204.85 | 1114.80 ±264.93 | 1289.10 ±180.87 |
| RFDflx240˚/s (Nm/mS) | 392.70±94.47 | 372.10±145.84 | 461.60±233.78 | 441.50±192.00 | 386.90 ±107.80 | 449.70±186.28 | 512.00±233.74 | 525.10±186.46 |
| PTQext120˚/s (Nm) | 224.90±21.00 | 235.20±20.92 | 216.10±26.18 | 220.80±36.26 | 226.40±18.26 | 243.10±26.46 | 227.90±32.08 | 241.40±37.19 |
| PTQflx120˚/s (Nm) | 115.80±11.63 | 114.60±15.26 | 109.40±21.13 | 125.10±11.08 | 116.60±13.09 | 120.30±14.23 | 118.90±15.65 | 131.80±14.11 |
| PTQext240˚/s (Nm) | 167.60±18.46 | 170.80±13.82 | 160.40±27.25 | 182.70±40.21 | 166.80±19.99 | 183.10±21.96 | 169.90±28.84 | 191.20±34.48 |
| PTQflx240˚/s (Nm) | 97.10±8.64 | 91.80±10.65 | 93.80±13.99 | 101.10±14.01 | 96.90±11.79 | 104.40±13.35 | 100.20±14.45 | 112.20±12.88 |
| RPText120˚/s (%) | 277.20±26.43 | 294.70±32.46 | 291.90±40.78 | 279.60±53.32 | 279.10±21.55 | 304.60±34.06 | 306.90±40.58 | 303.20±40.27 |
| RPTflx120˚/s (%) | 142.30±10.67 | 143.80±21.29 | 147.00±25.10 | 157.70±17.52 | 143.90±16.62 | 150.30±16.19 | 159.90±18.22 | 165.60±14.30 |
| RPText240˚/s (%) | 206.30±19.17 | 214.90±27.71 | 216.00±34.44 | 229.90±51.07 | 205.00±16.79 | 228.90±25.41 | 229.10±38.66 | 238.50±23.57 |
| RPTflx240˚/s (%) | 119.70±10.87 | 114.90±14.24 | 126.40±17.43 | 127.70±21.13 | 119.70±16.68 | 130.20±14.49 | 134.70±17.22 | 140.80±11.42 |
| APext120˚/s (watts) | 279.80±36.21 | 296.80±51.35 | 273.10±23.38 | 286.10±37.05 | 287.10±31.19 | 298.00±39.84 | 291.30±23.59 | 306.70±43.15 |
| APflx120˚/s (watts) | 152.40±28.17 | 155.40±30.69 | 152.40±17.65 | 162.60±17.63 | 154.60±26.55 | 168.70±29.93 | 167.20±16.72 | 181.20±18.26 |
| APext240˚/s (watts) | 345.60±49.46 | 358.80±48.78 | 329.20±28.82 | 362.70±61.81 | 352.60±36.87 | 363.50±70.50 | 355.10±30.03 | 383.10±79.22 |
| APflx240˚/s (watts) | 186.50±44.13 | 177.10±41.78 | 194.50±26.10 | 200.40±9.91 | 177.80±29.86 | 193.30±41.75 | 199.20±26.19 | 222.50±24.01 |

PT: plyometric training, PTS: plyometric training with speed overload, PTW: plyometric training with weight overload, PTSW: plyometric training with speed and weight overload, SJH: Sarjent's jump height, SPJH: Spike jump height, ShT: Sheppard test, ext: Extension, flx: Flexion, MVIC: Muscle voluntary isometric contraction, RFD: Average rate of force development, PTQ: Absolute peak torque, RPT: Relative peak torque, AP: Average power, s: Second, cm: Centimeter, Nm: Newton meters.

For the highest jump of ShT, a significant main effect was observed ($F_{1.00}$ = 50.17, P = 0.001, $pEta^2$ = 0.582) (Table 4). Post-hoc tests showed substantial improvements in PTS (P = 0.001), PTW (P = 0.001), and PTSW (P = 0.001) in the post-test compared to the pre-test. In the post-test, PTSW significantly outperformed PT (P = 0.035), while no other between-group differences were observed. Pre-test results showed no significant differences among groups (P > 0.05) (Fig 4).

No significant main effects were found for the average time of ShT ($F_{1.00}$ = 0.05, P = 0.815, $pEta^2$ = 0.002) or the shortest time of ShT ($F_{1.00}$ = 0.20, P = 0.655, $pEta^2$ = 0.006) (Table 4). Additionally, no considerable differences were observed between groups for the average time of ShT or the shortest time of ShT in either pre-test or post-test (P > 0.05) (Fig 4).

**Table 4. Comparison of the pre-test and post-test in each group.**

| | | | PT (n = 10) | PTS (n = 10) | PTW (n = 10) | PTSW (n = 10) |
|---|---|---|---|---|---|---|
| | | | Post | | | |
| Agility (s) | MD | Pre | -0.01 | -0.27 | -0.14 | -0.19 |
| | Sig | | 0.908 | 0.007 | 0.143 | 0.047 |
| | 95%CI | | -0.20–0.18 | -0.46–-0.07 | -0.33–0.05 | -0.38–-0.01 |
| SJH (cm) | MD | Pre | 0.10 | 1.70 | 1.30 | 3.20 |
| | Sig | | 0.887 | 0.012 | 0.041 | 0.001 |
| | 95%CI | | -1.20–1.40 | 0.39–3.00 | 0.01–2.60 | 1.89–4.50 |
| SPJH (cm) | MD | Pre | -0.20 | 1.90 | 2.30 | 3.40 |
| | Sig | | 0.732 | 0.002 | 0.001 | 0.001 |
| | 95%CI | | -1.37–0.97 | 0.72–3.07 | 1.12–3.47 | 2.22–4.57 |
| Average time of ShT (s) | MD | Pre | -0.01 | 0.00 | -0.08 | 0.05 |
| | Sig | | 0.911 | 0.991 | 0.329 | 0.540 |
| | 95%CI | | -0.19–0.17 | -0.17–0.18 | -0.26–0.09 | -0.12–0.23 |
| Shortest time of ShT (s) | MD | Pre | -0.06 | 0.00 | -0.08 | 0.07 |
| | Sig | | 0.381 | 0.906 | 0.252 | 0.309 |
| | 95%CI | | -0.22–0.08 | -0.14–0.16 | -0.24–0.06 | -0.07–0.23 |
| Average jump height of ShT (cm) | MD | Pre | -0.10 | 2.10 | 1.80 | 3.40 |
| | Sig | | 0.862 | 0.001 | 0.003 | 0.001 |
| | 95%CI | | -0.506–0.306 | 1.063–3.136 | 0.459–3.140 | 1.486–5.313 |
| Highest jump of ShT (cm) | MD | Pre | 0.00 | 1.90 | 2.20 | 3.40 |
| | Sig | | 1.000 | 0.001 | 0.001 | 0.001 |
| | 95%CI | | -1.07–1.07 | 0.82–2.97 | 1.12–3.27 | 2.32–4.47 |
| MVICext45˚ (Nm) | MD | Pre | -0.60 | 6.32 | 10.10 | 21.95 |
| | Sig | | 0.929 | 0.349 | 0.138 | 0.002 |
| | 95%CI | | -14.11–12.91 | -7.19–19.83 | -3.41–23.61 | 8.43–35.46 |
| MVICflx45˚ (Nm) | MD | Pre | -1.50 | 3.20 | 10.80 | 8.05 |
| | Sig | | 0.717 | 0.440 | 0.012 | 0.047 |
| | 95%CI | | -9.82–6.82 | -5.12–11.52 | 2.48–19.12 | 0.27–16.37 |
| RFDext120˚/s (Nm/mS) | MD | Pre | -1.00 | 130.00 | 110.20 | 125.60 |
| | Sig | | 0.981 | 0.003 | 0.012 | 0.005 |
| | 95%CI | | -85.30–83.30 | 45.69–214.30 | 25.89–194.50 | 41.29–209.90 |
| RFDflx120˚/s (Nm/mS) | MD | Pre | 19.10 | 26.70 | 89.90 | 98.20 |
| | Sig | | 0.654 | 0.531 | 0.040 | 0.026 |
| | 95%CI | | -66.48–104.68 | -58.88–112.28 | 4.31–175.48 | 12.61–183.78 |
| RFDext240˚/s (Nm/mS) | MD | Pre | -32.20 | 142.00 | 59.90 | 196.80 |
| | Sig | | 0.383 | 0.001 | 0.109 | 0.001 |
| | 95%CI | | -106.12–41.72 | 68.07–215.92 | -14.02–133.82 | 122.87–270.72 |
| RFDflx240˚/s (Nm/mS) | MD | Pre | -5.80 | 77.60 | 50.40 | 83.60 |
| | Sig | | 0.860 | 0.023 | 0.131 | 0.015 |
| | 95%CI | | -71.92–60.32 | 11.47–143.72 | -15.72–116.52 | 17.47–149.72 |
| PTQext120˚/s (Nm) | MD | Pre | 1.50 | 7.90 | 11.80 | 20.60 |
| | Sig | | 0.634 | 0.272 | 0.105 | 0.006 |
| | 95%CI | | -12.87–15.87 | -6.47–22.27 | -2.57–26.17 | 6.22–34.97 |
| PTQflx120˚/s (Nm) | MD | Pre | 0.80 | 5.70 | 9.50 | 6.70 |
| | Sig | | 0.823 | 0.118 | 0.011 | 0.068 |
| | 95%CI | | -6.41–8.01 | -1.51–12.91 | 2.28–16.71 | -0.51–13.91 |

(*Continued*)

**Table 4.** (Continued)

|  |  |  | PT (n = 10) | PTS (n = 10) | PTW (n = 10) | PTSW (n = 10) |
|---|---|---|---|---|---|---|
|  |  |  | Post |  |  |  |
| **PTQext240˚/s (Nm)** | **MD** | **Pre** | -0.80 | 12.30 | 9.50 | 8.50 |
|  | **Sig** |  | 0.909 | 0.087 | 0.182 | 0.231 |
|  | **95%CI** |  | -14.96–13.36 | -1.86–26.46 | -4.66–23.66 | -5.66–22.66 |
| **PTQflx240˚/s (Nm)** | **MD** | **Pre** | -0.20 | 12.60 | 6.40 | 11.10 |
|  | **Sig** |  | 0.955 | 0.001 | 0.080 | 0.003 |
|  | **95%CI** |  | -7.39–6.99 | 5.40–19.79 | -0.79–13.59 | 3.90–18.29 |
| **RPText120˚/s (%)** | **MD** | **Pre** | 1.90 | 9.90 | 15.00 | 23.60 |
|  | **Sig** |  | 0.818 | 0.236 | 0.076 | 0.007 |
|  | **95%CI** |  | -14.74–18.54 | -6.74–26.54 | -1.64–31.64 | 6.95–40.24 |
| **RPTflx120˚/s (%)** | **MD** | **Pre** | 1.60 | 6.50 | 12.90 | 7.90 |
|  | **Sig** |  | 0.717 | 0.147 | 0.006 | 0.080 |
|  | **95%CI** |  | -7.29–10.49 | -2.39–15.39 | 4.00–21.79 | -0.99–16.79 |
| **RPText240˚/s (%)** | **MD** | **Pre** | -1.30 | 14.00 | 13.10 | 8.60 |
|  | **Sig** |  | 0.882 | 0.117 | 0.141 | 0.330 |
|  | **95%CI** |  | -18.96–16.36 | -3.66–31.66 | -4.56–30.76 | -9.06–26.26 |
| **RPTflx240˚/s (%)** | **MD** | **Pre** | 0.00 | 15.30 | 8.30 | 13.10 |
|  | **Sig** |  | 1.000 | 0.002 | 0.073 | 0.006 |
|  | **95%CI** |  | -9.13–9.13 | 6.16–24.43 | -0.83–17.43 | 3.996–22.23 |
| **APext120˚/s (watts)** | **MD** | **Pre** | 7.30 | 1.20 | 18.20 | 20.60 |
|  | **Sig** |  | 0.210 | 0.835 | 0.003 | 0.001 |
|  | **95%CI** |  | -4.29–18.89 | -10.39–12.79 | 6.60–29.79 | 9.00–32.19 |
| **APflx120˚/s (watts)** | **MD** | **Pre** | 2.20 | 13.30 | 14.80 | 18.60 |
|  | **Sig** |  | 0.759 | 0.070 | 0.045 | 0.013 |
|  | **95%CI** |  | -12.23–16.63 | -1.13–27.73 | 0.36–29.23 | 4.16–33.03 |
| **APext240˚/s (watts)** | **MD** | **Pre** | 7.00 | 4.70 | 25.90 | 20.40 |
|  | **Sig** |  | 0.503 | 0.653 | 0.017 | 0.047 |
|  | **95%CI** |  | -13.99–27.99 | -16.29–25.69 | 4.90–46.89 | 0.59–41.39 |
| **APflx240˚/s (watts)** | **MD** | **Pre** | -8.70 | 16.20 | 4.70 | 22.10 |
|  | **Sig** |  | 0.422 | 0.139 | 0.663 | 0.066 |
|  | **95%CI** |  | -30.41–13.01 | -5.51–37.91 | -17.01–26.41 | -0.38–43.81 |

PT: plyometric training, PTS: plyometric training with speed overload, PTW: plyometric training with weight overload, PTSW: plyometric training with speed and weight overload, SJH: Sarjent's jump height, SPJH: Spike jump height, ShT: Sheppard test, ext: Extension, flx: Flexion, MVIC: Muscle voluntary isometric contraction, RFD: Average rate of force development, PTQ: Absolute peak torque, RPT: Relative peak torque, AP: Average power, s: Second, cm: Centimeter, Nm: Newton meters, MD: Mean Difference, CI: Confidence Interval.

## 3.2 Isometric and isokinetic strength test results

The repeated measure analysis test results showed that the main effect on MVICext45˚ was significant ($F_{1.00}$ = 8.03, P = 0.007, pEta$^2$ = 0.182). Also, the Bonferroni test indicated that the MVICext45˚ in PTSW (P = 0.002) was substantially higher in the post-test compared to the pre-test; however, no significant difference was observed between the post-test and pre-test in the PT (P = 0.929), PTS (P = 0.349), and PTW (P = 0.138) (Table 4). In addition, there was no considerable difference between the studied groups in the pre-test (P>0.05) and post-test (P>0.05) (Fig 5).

Statistical analysis of data showed that the main effect on the MVICflx45˚ was significant ($F_{1.00}$ = 6.27, P = 0.017, pEta$^2$ = 0.148), and the results of the Bonferroni test indicated that

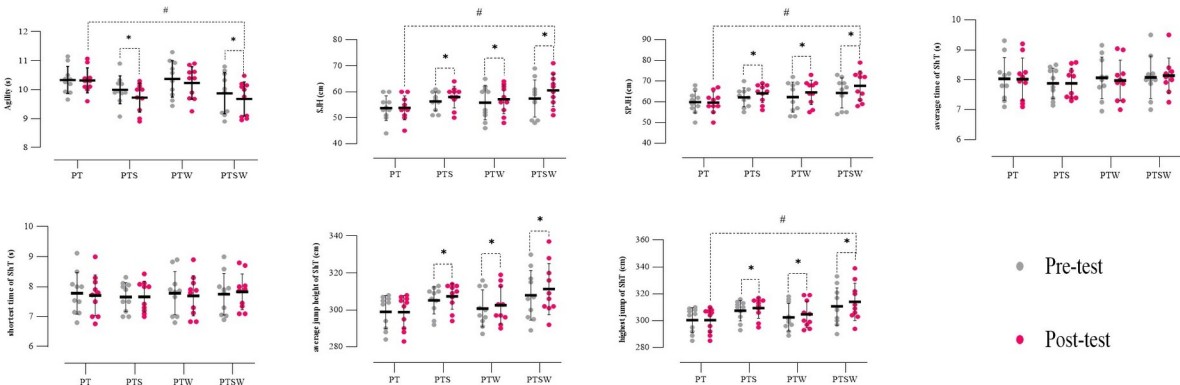

**Fig 4. Individual responses, means, and standard deviation of the functional tests results in the four groups.** PT: plyometric training, PTS: plyometric training with speed overload, PTW: plyometric training with weight overload, PTSW: plyometric training with speed and weight overload, SJH: Sarjent's jump height, SPJH: Spike jump height, ShT: Sheppard test, s: Second, cm: Centimeter. *: significant difference compared to the pre-test. #: significant difference of D of groups compared to D of PT.

MVICflx45° in the PTW (P = 0.012), and PTSW (P = 0.047) improved significantly in the post-test compared to the pre-test; however, no significant difference was observed between the post-test and pre-test in the PT (P = 0.717) and PTS (P = 0.440) (Table 4). Additionally, there were no significant differences between the studied groups in the pre-test (P>0.05) and post-test (P>0.05) (Fig 5).

According to the results of the analysis, the main effect on the RFDext120°/s was considerable ($F_{1.00}$ = 19.25, P = 0.001, pEta$^2$ = 0.348). In addition, the results of the post-hoc test indicated that the RFDext120°/s in PTS (P = 0.003), PTW (P = 0.012), and PTSW (P = 0.005) increased significantly in the post-test compared to the pre-test. Nevertheless, no significant difference was observed between the post-test and pre-test in the PT (P = 0.981) (Table 4). Moreover, the Bonferroni test showed that in the post-test, the RFDext120°/s improved substantially in PTS (P = 0.048), and PTSW (P = 0.033) compared to the PT, but no significant differences were observed between PTW and PT (P = 1.000), and PTS and PTW (P = 0.535). Also, there was no considerable difference between the studied groups in the pre-test (P>0.05) (Fig 5).

The repeated measure analysis test results showed that the main effect on the RFDflx120°/s was significant ($F_{1.00}$ = 7.68, P = 0.009, pEta$^2$ = 0.176). The Bonferroni test indicated that the RFDflx120°/s in PTW (P = 0.040), and PTSW (P = 0.026) were substantially higher in the post-test compared to the pre-test; however, no significant difference was observed between the post-test and pre-test in the PT (P = 0.654), and PTS (P = 0.531) (Table 4). In addition, there was no considerable difference between the studied groups in the pre-test (P>0.05) and post-test (P>0.05) (Fig 5).

Statistical data analysis showed that the main effect on the RFDext240°/s was significant ($F_{1.00}$ = 25.27, P = 0.001, pEta$^2$ = 0.413). The results of the Bonferroni test indicated that RFDext240°/s in the PTS (P = 0.001), and PTSW (P = 0.001) improved significantly in the post-test compared to the pre-test; however, no significant difference was observed between the post-test and pre-test in the PT (P = 0.383) and PTW (P = 0.109) (Table 4). Additionally, the post-hoc test showed that in the post-test, the RFDext240°/s improved significantly in PTS (P = 0.008) and PTSW (P = 0.010) compared to the PT, but no significant differences were observed between PTW and PT (P = 0.814), and PTS and PTW (P = 0.352). Also, there were no significant differences between the studied groups in the pre-test (P>0.05) (Fig 5).

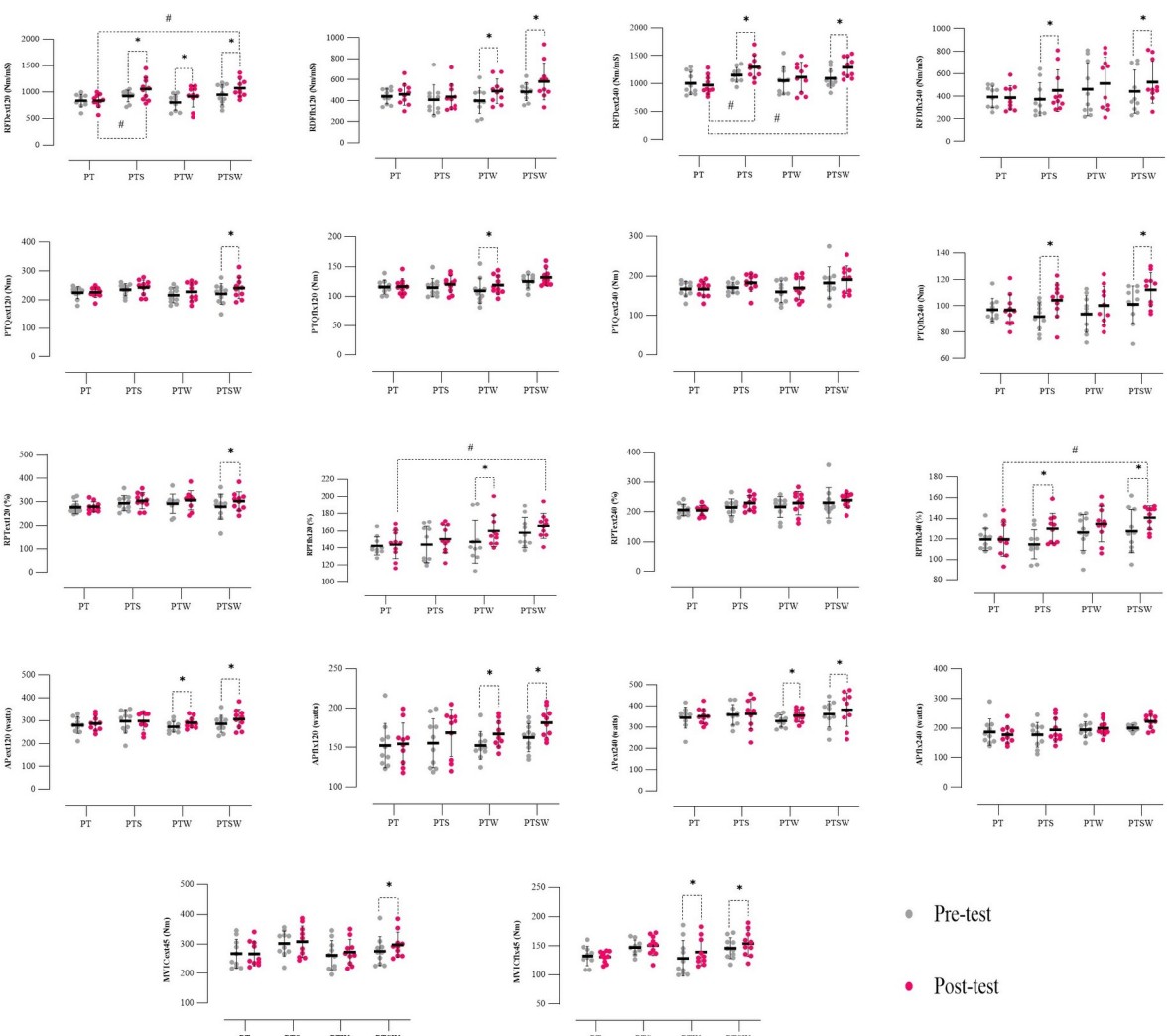

**Fig 5. Individual responses, means, and standard deviation of the isometric and isokinetic parameters in the four groups.** PT: plyometric training, PTS: plyometric training with speed overload, PTW: plyometric training with weight overload, PTSW: plyometric training with speed and weight overload, ext: Extension, flx: Flexion, MVIC: Muscle voluntary isometric contraction, RFD: Average rate of force development, PTQ: Absolute peak torque, RPT: Relative peak torque, AP: Average power, s: Second, Nm: Newton meters.

According to the results, the main effect on the RFDflx240°/s was considerable ($F_{1.00}$ = 9.96, P = 0.003, pEta$^2$ = 0.217). In addition, the results of the post-hoc test indicated that the RFDflx240°/s in PTS (P = 0.023) and PTSW (P = 0.015) increased significantly in the post-test compared to the pre-test. Nevertheless, no significant difference was observed between the post-test and pre-test in the PT (P = 0.860), and PTW (P = 0.131) (Table 4). Moreover, the Bonferroni test showed that there was no considerable difference between the studied groups in the pre-test (P>0.05) and post-test (P>0.05) (Fig 5).

The results of the repeated measure analysis test showed that the main effect on the PTQext120°/s was significant ($F_{1.00}$ = 8.69, P = 0.006, pEta$^2$ = 0.195), and the Bonferroni test indicated that the PTQext120°/s in PTSW (P = 0.006) was substantially more in the post-test compared to the pre-test; however, no significant difference was observed between the post-test and pre-test in the PT (P = 0.834), PTS (P = 0.272), and PTW (P = 0.105) (Table 4). In

addition, there was no considerable difference between the studied groups in the pre-test (P>0.05) and post-test (P>0.05) (Fig 5).

Statistical analysis of data showed that the main effect on the PTQflx120˚/s was significant ($F_{1.00}$ = 10.17, P = 0.003, pEta$^2$ = 0.220). The results of the Bonferroni test indicated that the PTQflx120˚/s in the PTW (P = 0.011) improved significantly in the post-test compared to the pre-test, but no significant difference was observed between the post-test and pre-test in the PT (P = 0.823), PTS (P = 0.118), and PTSW (P = 0.068) (Table 4). Also, there were no significant differences between the studied groups in the pre-test (P>0.05) and post-test (P>0.05) (Fig 5).

According to the results of the analysis, the main effect on the PTQflx240˚/s was considerable ($F_{1.00}$ = 17.74, P = 0.001, pEta$^2$ = 0.330). In addition, the results of the post-hoc test indicated that the PTQflx240˚/s in PTS (P = 0.001) and PTSW (P = 0.003) increased significantly in the post-test compared to the pre-test. However, no significant difference was observed between the post-test and pre-test in the PT (P = 0.955) and PTW (P = 0.080) (Table 4). Moreover, the Bonferroni test showed that there was no considerable difference between the studied groups in the pre-test (P>0.05) and post-test (P>0.05) (Fig 5).

The results of the repeated measure analysis test showed that the main effect on the RPText120˚/s was significant ($F_{1.00}$ = 36.00, P = 0.004, pEta$^2$ = 0.207), and the Bonferroni test indicated that the RPText120˚/s in PTSW (P = 0.007) was substantially higher in the post-test compared to the pre-test, but, no significant difference was observed between the post-test and pre-test in the PT (P = 0.818), PTS (P = 0.236), and PTW (P = 0.076) (Table 4). In addition, there was no substantial difference between the groups of the study in the pre-test (P>0.05) and post-test (P>0.05) (Fig 5).

Statistical analysis of data showed that the main effect on the RPTflx120˚/s was significant ($F_{1.00}$ = 10.84, P = 0.002, pEta$^2$ = 0.232). The results of the post-hoc test showed that the RPTflx120˚/s in the PTW (P = 0.006) increased considerably in the post-test compared to the pre-test, but no significant difference was observed between the post-test and pre-test in the PT (P = 0.717), PTS (P = 0.147), and PTSW (P = 0.080) (Table 4). Additionally, the Bonferroni test indicated that in the post-test, the RPTflx120˚/s improved significantly in PTSW (P = 0.033) compared to the PT. However, no significant differences were observed between PTS and PT (P = 1.000), PTW and PT (P = 0.214), and PTS and PTW (P = 1.000). Also, there was no significant difference between the studied groups in the pre-test (P>0.05) (Fig 5).

According to the results, the main effect on the RPTflx240˚/s was considerable ($F_{1.00}$ = 36.00, P = 0.001, pEta$^2$ = 0.316). In addition, the results of the post-hoc test indicated that the RPTflx240˚/s in PTS (P = 0.002) and PTSW (P = 0.006) improved significantly in the post-test compared to the pre-test. Nevertheless, no significant difference was observed between the post-test and pre-test in the PT (P = 1.000), and PTW (P = 0.073) (Table 4). Moreover, the Bonferroni test showed that in the post-test, the RPTflx240˚/s was significantly higher in PTSW (P = 0.021) compared to the PT, but no substantial difference was observed between PTS and PT (P = 0.777), PTW and PT (P = 0.198), and PTS and PTW (P = 1.000). Also, there was no significant difference between the studied groups in the pre-test (P>0.05) (Fig 5).

The results of the analysis showed that the main effect on the APext120˚/s was significant ($F_{1.00}$ = 17.10, P = 0.001, pEta$^2$ = 0.322), and the Bonferroni test indicated that the APext120˚/s in PTW (P = 0.003) and PTSW (P = 0.001) was substantially more in the post-test compared to the pre-test; however, no significant difference was observed between the post-test and pre-test in the PT (P = 0.210) and PTS (P = 0.835) (Table 4). In addition, there was no considerable difference between the groups of the study in the pre-test (P>0.05) and post-test (P>0.05) (Fig 5).

Statistical analysis of data showed that the main effect on the APflx120˚/s was considerable ($F_{1.00}$ = 11.80, P = 0.002, pEta$^2$ = 0.247). The results of the Bonferroni test indicated that the

APflx120°/s in the PTW (P = 0.045) and PTSW (P = 0.013) improved significantly in the post-test compared to the pre-test, but no significant difference was observed between the post-test and pre-test in the PT (P = 0.759) and PTS (P = 0.070) (Table 4). Also, there were no significant differences between the studied groups in the pre-test (P>0.05) and post-test (P>0.05) (Fig 5).

According to the analysis results, the main effect on the APext240°/s was considerable ($F_{1.00}$ = 7.84, P = 0.008, pEta$^2$ = 0.179). In addition, the post-hoc test results indicated that the APext240°/s in PTW (P = 0.017) and PTSW (P = 0.047) increased significantly in the post-test compared to the pre-test. However, no significant difference was observed between the post-test and pre-test in the PT (P = 0.503) and PTS (P = 0.653) (Table 4). Moreover, the Bonferroni test showed that there was no considerable difference between the studied groups in the pre-test (P>0.05) and post-test (P>0.05) (Fig 5).

The results showed that the main effect was not significant on the PTQext240°/s ($F_{1.00}$ = 4.46, P = 0.062, pEta$^2$ = 0.110), the RPText240°/s ($F_{1.00}$ = 3.89, P = 0.056, pEta$^2$ = 0.098), and the APflx240°/s ($F_{1.00}$ = 2.56, P = 0.118, pEta$^2$ = 0.067) (Table 4). Also, no considerable difference was observed for the PTQext240°/s (P>0.05), the RPText240°/s (P>0.05), and the APflx240°/s (P>0.05), between the studied groups in the pre-test and post-test (Fig 5).

## 4. Discussion

This study evaluated the impact of speed and weight overload on volleyball players' isokinetic strength, explosive power, and agility. Although some research has been conducted, the effects of speed and weight overload training on volleyball players' performance remain uncertain and contradictory. We hypothesize that overspeed training can enhance the power of the lower limbs and improve jumping ability by reducing a jumper's effective mass [12]. Consequently, it can also increase the peak acceleration during the jump due to reduced load conditions [13]. On the other hand, the overload principle states that disruption of the body's homeostasis, including cells, tissues, and organs, is necessary for effective adaptation to exercise, so that overload training can be efficient [41]. In summary, the results of the present study indicated that four weeks of PTS and PTW can significantly improve jumping ability and isometric and isokinetic strength parameters in male volleyball players. However, it seems that the combination of these two types of training (PTSW) had a better effect on these players' jumping performance and isometric and isokinetic strength.

Consistent with these findings, several studies have also shown that PTS and PTW interventions positively affect jumping performance and isokinetic muscle strength [42–48]. In 2011, the effects of assisted jumping on jump height in elite athletes were investigated, and assisted jumping training resulted in a 2.7 cm increase in cross jump and a 2.6 cm increase in spike jump [7]. Also, the findings of this study align with previous research, which showed significant improvements in isokinetic strength and jumping ability following plyometric training with additional loads in young athletes [43]. These results further emphasize the effectiveness of such training protocols in enhancing athletic performance. However, some studies have also reported inconsistent results [44–46]. Stien et al. (2020) investigated the effect of plyometric training with excessive speed or overload on jump height, and the results showed that no significant difference was found in the jump between groups in the training group with excessive speed [47]. Markovich et al. reported that eight weeks of plyometric training (with different types and sizes of external loads) improved jump performance compared to a control group without significantly affecting maximal squat strength (1RM) [48]. Additionally, Ioannides et al. [49] reported that plyometric training improved horizontal jump performance and maximal isometric strength without changing RFD parameters. However, differences in

training protocols and RFD assessments, as well as the smaller sample size in the previous study (n = 12 vs n = 40), may have contributed to the discrepancy in results. Additionally, Wilson et al. [48] reported that 5–10 weeks of deep jump training did not significantly affect isokinetic performance in strength athletes. It is worth noting that the Wilson study differed from the present study. For instance, the training volume was only 60 jumps per week, whereas it was 248–372 jumps per week in the present study. The previous study used only one exercise, while the present used four. Lastly, the assessment methods used in the last study differed from those used in the present study. Therefore, the differences in the results obtained can be attributed to these methodological differences.

According to Sheppard, using an assisted jump technique can increase the contraction speed in leg and trunk extensor muscles, improving jump height [7]. Additionally, stimulating deep body receptors like muscle spindles, Golgi tendon organs (GTO), and mechanoreceptors in joint capsules and ligaments can affect the agonist and antagonist muscles. When the muscle spindle is stretched, efferent nerve firing increases and the message's strength depends on the stretch's speed. The faster the stretch, the stronger the neural signal sent from the muscle spindle, resulting in a more significant muscle contraction or shortening cycle of the plyometric movement [50, 51]. Therefore, the increase in height of different jumps and isokinetic strength parameters observed in the present study can be attributed to the rise in the speed of nerve message transmission and more robust muscle contractions.

In addition, previous studies have shown that the isokinetic strength of knee extensors [52, 53] and knee flexors [54, 55] significantly correlates with jumping ability. The results of the study showed significant improvements in PTQflx240˚/s, and RPTflx240˚/s for the PTS group, PTQflx120˚/s, RPTflx120˚/s, APext120˚/s, APflx120˚/s, and APext240˚/s for the PTW group, and PTQext120˚/s, PTQext240˚/s, RPText120˚/s, RPTflx240˚/s, APext120˚/s, APflx120˚/s, and APext240˚/s for the PTSW group after the intervention. These improvements may contribute to enhancing jumping performance. For example, Kubo et al. reported that after 12 weeks of plyometric training, jumping ability increased with significant improvement in strength and activation of plantar flexors [56]. In another study, Malisoux et al. showed that after eight weeks of plyometric training, vertical jump performance increased significantly (13%) with an increase in 1RM leg press (12%) [57]. Also, in a systematic review, Nishiumi et al. (2023) suggested that improving the eccentric strength of lower limb muscles is related to increasing jumping ability [58]. Therefore, after interventions, the improvement in jumping ability may be due to improved isokinetic strength of knee extensors and flexors.

Research suggests that there are different mechanisms responsible for improvements in strength, power, and jumping ability after performing specific exercises. Increased muscle hypertrophy and neural drive to agonist muscles may be the primary factors contributing to increased strength and energy after performing particular exercises such as PTS and PTW [59]. On the other hand, changes in the mechanical properties of the muscle-tendon complex, such as maximum tendon elongation and tendon elastic energy, may be responsible for the improved jumping ability after PTSW [60]. Moreover, wearing weighted vests during exercises can enhance performance by strengthening antigravity muscles, increasing vertical propulsion, and reducing contact time [61]. Hyper-gravity training (carrying additional weight) significantly stresses antigravity muscle groups, improving repetitive jump performance in the leg, knee, and ankle extensor muscles [62]. Studies have demonstrated that wearing a weighted vest (11% of body weight) for three weeks can shift the force-velocity curve to the right, improving power output during jumps [63]. These findings highlight the benefits of performing exercises with weighted vests to improve jumping ability and strength-based activities [64].

According to the present study, MVIC and RFD improvements can also improve jumping performance. RFD is a parameter that measures how quickly an athlete can produce peak

force and is calculated with a force-time curve [65]. Previous studies have shown that increased RFD is associated with improved jumping ability after plyometric training [60, 66–68]. Therefore, the improvement in jumping ability after the interventions can be attributed to increases in RFDext120°/s, RFDext240°/s, and RFDflx240°/s for the PTS group, MVICflx45° and RFDflx120°/sfor the PTW group, and MVICext45°, MVICflx45°, RFDext120°/s, RFDflx120°/s, RFDext240°/s and RFDflx240°/s for the PTSW group. Also, improvements in MVIC and RFD can be explained by neuromuscular adaptations stimulated by the training protocol. Factors such as motor unit recruitment and discharge rate [69, 70], muscle fiber type composition [57, 66, 70], muscle-tendon architecture [56, 57, 70], $Ca^{2+}$ sensitivity and contractility [57, 70, 71] affect RFD [72]. Previous studies have shown that plyometric training increases neural drive, motor unit recruitment, isometric maximal voluntary contraction, and RFD in knee extensors [69, 73] and plantar flexors [74], leading to increased jumping performance [69, 73]. Plyometric exercises (five weeks of jump/sprint training) can also decrease fascicle angle and increase fascicle length in knee extensors [75]. Additionally, after eight weeks of plyometric training, peak force, cross-sectional area, and maximum shortening velocity ($V_0$) in type I, IIa, and IIx fibers have been shown to increase significantly [57]. The increase in $V_0$ of muscle fibers (type I and II fibers) is likely due to the nature of PTS and PTW and high-intensity contractions that recruit all fiber types [57, 70].

The results of the current study revealed that the average time of ShT, and shortest time of ShT had no significant increases after the interventions in all studied groups. Consistent with these findings, Brughelli et al. showed that conventional training programs that include gravity-dependent weight exercises performed in a vertical direction, such as Olympic-style lifts and plyometric squats, have failed to induce the change of direction (COD) [76]. In contrast, the results of previous studies have revealed better physiological responses and more significant stimuli offered by iso-inertial training when compared to traditional weight training [77, 78]. For example, Fiorilli et al. showed that six weeks of iso-inertial eccentric overload training improves the COD of soccer players [77]. A faster stretching phase increases the storage and release of energy available at the end of the eccentric phase, allowing an improvement of kinetic energy seen in movement velocity [79]. During COD, an athlete needs eccentric force to rapidly decelerate and concentric strength to accelerate in a new direction [80]. It has been demonstrated that eccentric strength stimulates the addition of sarcomeres in series, which increases the muscle fascicle length [81]. Regarding performance, greater tendon stiffness is critical in rapidly transmitting strength from the muscles to the skeletal system [82]. The results of the present study showed an improvement in the agility of the participants, while it was expected that the PTS and PTW protocols could positively impact average time of ShT, and shortest time of ShT, these results were not observed in the current study, possibly due to differences in training protocols, RFD assessments, exercise placement, and assessment methods. Although, increasing $V_0$ of muscle fibers (type I and II fibers) due to the nature of PTS and PTW and high-intensity contractions that recruit all fiber types, might be caused the improvement in the agility in the PTS and PTSW groups [57, 70].

Despite the promising findings of this study, some limitations should be acknowledged. Firstly, the short duration of the intervention (four weeks) limits the generalization of the findings to long-term training adaptations. Future studies should consider extended training periods to examine sustained effects and potential plateau phases. Secondly, the study included only male volleyball players, which restricts the applicability of the results to female athletes or players from other sports disciplines. Including diverse athlete profiles in subsequent research could provide more comprehensive insights. Thirdly, the study relied on specific plyometric training protocols with speed and weight overloads, which may not fully represent the diversity of training methods available. Exploring other variations of overload training could

enhance the understanding of optimal methods for performance improvements. Finally, environmental and psychological factors, such as motivation and external conditions during testing, were not controlled, which might have influenced the results. Addressing these factors in future research will help validate and refine the conclusions of this study.

## 5. Conclusion

In conclusion, the present study demonstrated that incorporating speed and weight overloads into plyometric training (PTSW) effectively enhances isokinetic strength, explosive power, and jumping performance in male volleyball players. The combination of speed and weight overloads showed the most significant improvements across various performance metrics, outperforming speed or weight overloads alone. These findings suggest that volleyball athletes can benefit from integrating PTSW into their training regimens to optimize their physical capabilities and competitive performance.

## Supporting information

**S1 File. Raw data in excel worksheet.**
(XLSX)

## Author Contributions

**Conceptualization:** Hamed Esmaeili.

**Data curation:** Ahmad Reza Iranpour, Babak Imanian.

**Formal analysis:** Ahmad Reza Iranpour.

**Investigation:** Ahmad Reza Iranpour.

**Methodology:** Javad Nemati.

**Project administration:** Mohammad Hemmatinafar.

**Software:** Mohsen Salesi, Babak Imanian.

**Supervision:** Mohammad Hemmatinafar.

**Validation:** Javad Nemati.

**Visualization:** Mohsen Salesi, Hamed Esmaeili.

**Writing – original draft:** Ahmad Reza Iranpour, Mohammad Hemmatinafar, Javad Nemati, Mohsen Salesi, Hamed Esmaeili, Babak Imanian.

**Writing – review & editing:** Mohammad Hemmatinafar, Babak Imanian.

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
