## [Decision Letter · Decision Letter 0]

18 Nov 2024

PONE-D-24-43876The effects of plyometric training with speed and weight overloads on volleyball players' strength, power, and jumping performancePLOS ONE

Dear Dr. hemmatinafar,

Thank you for submitting your manuscript to PLOS ONE. After careful consideration, we feel that it has merit but does not fully meet PLOS ONE’s publication criteria as it currently stands. Therefore, we invite you to submit a revised version of the manuscript that addresses the points raised during the review process.

We look forward to receiving your revised manuscript.

Kind regards,

Julio Alejandro Henriques Castro da Costa

Academic Editor

PLOS ONE

**Journal Requirements:**

2. In the online submission form, you indicated that Data described in the manuscript will be available upon reasonable request.

3. We note that Figure 2 includes an image of a participant in the study. 

If you are unable to obtain consent from the subject of the photograph, you will need to remove the figure and any other textual identifying information or case descriptions for this individual."

Reviewers' comments:

Reviewer's Responses to Questions

**Comments to the Author**

1. Is the manuscript technically sound, and do the data support the conclusions?

Reviewer #1: Partly

Reviewer #2: Yes

2. Has the statistical analysis been performed appropriately and rigorously? 

Reviewer #1: No

Reviewer #2: Yes

3. Have the authors made all data underlying the findings in their manuscript fully available?

Reviewer #1: Yes

Reviewer #2: Yes

4. Is the manuscript presented in an intelligible fashion and written in standard English?

Reviewer #1: Yes

Reviewer #2: Yes

5. Review Comments to the Author

**Reviewer #1: **Comments :

This study underscores the role of gradual overload in boosting sports performance, particularly through plyometric training with speed and weight variations. It finds that combining these overloads significantly enhances jump height, power, and isokinetic strength in volleyball players, with the PTSW group achieving the greatest improvements. This suggests that adding speed and weight overload to plyometric training is an effective strategy for athletic performance gains.

The study is generally well conducted, but there are still parts that, in my opinion, present serious ambiguity and need to be revised.

Here are some comments:

General comments:

*Please refer to the PLOS ONE standard for authors following the instructions and guidelines: abstract format, references styles in the text, references, etc.

*Please add sentence/paragraph numbering at the beginning of each sentence.

*Some references are old. Please use more recent references.

Specific comments:

*Please add a brief description of the installation of the subjects, as technique and position are crucial in the assessment, both at isokinetic and isometric modes.

*The absolute peak torque is becoming increasingly rare to use in isokinetic assessments; we instead use the PT reported to body weight for greater accuracy. The Biodex provides this measure in the isokinetic parameter table. Try adding this parameter and comparing your results.

The use of the t-test in this study seems to me to be the main problem. Indeed, a repeated measures ANOVA should be considered instead of the t-test. The interactions between groups and parameters are also interesting to analyse and observe.

*Results and Discussion are well written, but it remains to be seen whether the statistics are correct, as these two sections are totally dependent on them.

**Reviewer #2:** Page 10:

1. One of the other methods of using the speed variable is overspeed training (OST). This method has been used for years as a particular exercise for athletics. Which is given to the athlete at a speed higher than the maximum speed by using tilt, … (consider the grouping into one single sentence)

2. this method is also called assisted Training. (suggestion: use quotation marks and no caps).

Page 12:

(Including full descriptions… : no capital letter needed.

There are several observations of this type in the same paragraph.

6. PLOS authors have the option to publish the peer review history of their article (what does this mean?). If published, this will include your full peer review and any attached files.

Reviewer #1: **Yes: **Nadhir Hammami

Reviewer #2: **Yes: **Diana Ciubotariu

---

## [Author Response · Author response to Decision Letter 0]

2 Dec 2024

We sincerely thank the reviewers for their thoughtful and constructive feedback on our manuscript. Your comments and suggestions have significantly enhanced the work's clarity, rigor, and overall quality. We appreciate the time and effort you dedicated to reviewing our submission and providing valuable insights, which allowed us to improve the manuscript and address essential aspects of the study. Your contributions are greatly appreciated. Please see the attached files. 

Best wishes,

Corresponding Author

---

## [Decision Letter · Decision Letter 1]

12 Dec 2024

The effects of plyometric training with speed and weight overloads on volleyball players' strength, power, and jumping performance

PONE-D-24-43876R1

Dear Dr. hemmatinafar,

We’re pleased to inform you that your manuscript has been judged scientifically suitable for publication and will be formally accepted for publication once it meets all outstanding technical requirements.

Kind regards,

Julio Alejandro Henriques Castro da Costa

Academic Editor

PLOS ONE

Additional Editor Comments (optional):

Reviewers' comments:

Reviewer's Responses to Questions

**Comments to the Author**

1. If the authors have adequately addressed your comments raised in a previous round of review and you feel that this manuscript is now acceptable for publication, you may indicate that here to bypass the “Comments to the Author” section, enter your conflict of interest statement in the “Confidential to Editor” section, and submit your "Accept" recommendation.

Reviewer #1: All comments have been addressed

2. Is the manuscript technically sound, and do the data support the conclusions?

Reviewer #1: Yes

3. Has the statistical analysis been performed appropriately and rigorously? 

Reviewer #1: Yes

4. Have the authors made all data underlying the findings in their manuscript fully available?

Reviewer #1: Yes

5. Is the manuscript presented in an intelligible fashion and written in standard English?

Reviewer #1: (No Response)

6. Review Comments to the Author

Reviewer #1: All comments are taken into account by the authors. I appreciate the work made, especially when statistical analysis was totally reworked. Good luck.

7. PLOS authors have the option to publish the peer review history of their article (what does this mean?). If published, this will include your full peer review and any attached files.

Reviewer #1: **Yes: **Nadhir Hammami

---

## [Editor Report · Acceptance letter]

23 Dec 2024

PONE-D-24-43876R1 

PLOS ONE

Dear Dr. hemmatinafar, 

I'm pleased to inform you that your manuscript has been deemed suitable for publication in PLOS ONE. Congratulations! Your manuscript is now being handed over to our production team.

Kind regards, 

on behalf of

Dr. Julio Alejandro Henriques Castro da Costa 

Academic Editor

PLOS ONE